# Relationship between the Void and Sound Absorption Characteristics of Epoxy Porous Asphalt Mixture Based on CT

**Xiaolong Li** [1,2]**, Junfeng Gao** [3,]*****, Hui Du** [2]**, Jingpeng Jia** [2]**, Xiaojie Zhao** [2] **and Tianqing Ling** [1]

1   School of Civil Engineering, Chongqing Jiaotong University, Chongqing 400074, China; xiaolongli1989@foxmail.com (X.L.); lingtq@163.com (T.L.)
2   Yunnan Research Institute of Highway Science and Technology, Kunming 650051, China; 18487301757@163.com (H.D.); jiajingpeng08@163.com (J.J.); zhaoxiaojie0810@126.com (X.Z.)
3   National & Local Joint Engineering Laboratory of Transportation and Civil Engineering Materials, Chongqing Jiaotong University, Chongqing 400074, China
*   Correspondence: jfgao@cqjtu.edu.cn; Tel.: +86-18069768936

**Abstract:** This study investigates the relationship between the void characteristics and sound absorption characteristics of an epoxy porous asphalt mixture. The specimens are scanned and reconstructed under different void fractions using X-ray computed tomography (CT) technology and digital image processing, and the sound absorption coefficients at different frequencies are obtained using an acoustic impedance tube. The relationship between void characteristics and sound absorption characteristics is analyzed using gray correlation. The test results exhibited a good correlation between the void characteristics of the epoxy porous asphalt mixture obtained by CT scanning (mesoscale) and the measured values (macroscale). The difference between the void fraction and connected void fraction gradually decreased with an increase in the void fraction. The relationship curve between the sound absorption coefficient and frequency exhibited a bimodal trend, and the peak value of the sound absorption coefficient increased with an increase in the void fraction. The order of the gray correlation degree between the peak and average values of the sound absorption coefficient and the void characteristic parameters is as follows: connected void fraction > void fraction > equivalent diameter of connected void > surface area of connected void > curvature.

**Keywords:** epoxy porous asphalt mixture; X-ray CT; void characteristics; sound absorption characteristics



## 1. Introduction

Porous asphalt pavement is widely used owing to its excellent properties, such as drainage, skid resistance, and noise reduction. As an environmentally friendly pavement, its excellent performance primarily comes from more connected voids in the structural layer [1,2]. Its noise reduction performance, which is its main function, is significantly affected by the size, morphology, distribution, and other characteristics of asphalt mixture voids.

Evaluation indexes based on volume characteristics, such as void fraction and connected void fraction, are typically used to evaluate the void characteristics measurement and its impact on the drainage and noise reduction performance of the asphalt mixture. Alvarez et al. [3] studied the internal structure of permeable friction course mixtures and assessed it in terms of air–void characteristics. Praticò et al. [4,5] observed that both measurement methodology and core diameter could substantially affect the specific gravity determination and air–void content estimate, and they proposed the theoretical derivation of models for some tests on asphalts by referring to the general theory for the percolation of water in rigid porous materials. Krishnan et al. [6] used the developed framework to model the movement of voids in asphalt concrete, proposed voids filled with asphalt (VFA) as a parameter to model bleeding of asphalt concrete, and studied its variation for different loading and mixture conditions. Xu et al. [7] analyzed the influence of void

characteristics on asphalt mixture drainage behavior and noted that void characteristics are the main factor that affects drainage capacity. Zhao et al. [8] remarked that asphalt mixture specimens under the same void fraction level exhibit different drainage characteristics, and the distribution and spatial trend of voids affected the drainage behavior. A study by Jiang et al. [9] demonstrated that different void equivalent diameters were produced at the same void fraction owing to different grading characteristics, which resulted in significant differences in noise reduction capacity. The distribution of voids in the asphalt mixture was non-uniform because it was a composite material. The void fraction only represented the overall size of the voids; however, it was difficult to characterize the complex meso morphological characteristics and spatial distribution of voids. Computed tomography (CT) technology provided a good entry point to better explain the void characteristics of asphalt mixture, owing to the advancement in digital image technology. Masad et al. [10,11] used X-ray CT scanning and image processing to study the content and distribution of voids in rotary compaction specimens and represented them using the Weibull model. Alvarez et al. [12] analyzed the vertical distribution of the void fraction in porous asphalt mixture using CT technology and established a prediction model for its permeability coefficient. In their study, Partl et al. [13] used CT technology to analyze the internal void distribution of three types of asphalt mixture specimens under different molding conditions and the difference between internal and external void fractions. Tan et al. [14,15] used industrial CT to reconstruct three-dimensional asphalt mixture specimens with different gradation types; the factors that affected void fraction test accuracy were examined using orthogonal experimental design and analysis of variance. Guo et al. [16] used X-ray CT technology to analyze the spatial distribution characteristics of voids in asphalt mixture specimens, and they demonstrated the non-uniformity of void distribution in space. Ahmad et al. [17] studied the three-dimensional structure of voids extracted using X-ray CT and proposed the solid content, circularity, and average void area as the evaluation indexes of the void section. Norhidayah et al. [18] analyzed the influence of mixture gradation characteristics on avoid section using X-ray CT technology. They reported that the voids formed by gradation with a coarser aggregate were rounder and more conducive to the formation of connected voids. Zhao et al. [19] divided the void structure into an effective void and invalid void according to its relative position to the drainage path, thereby distinguishing the different functions of different void structures in the process of drainage behavior. Pei et al. [20] investigated the influence of mineral aggregate gradation on the void characteristics in the internal structure of a porous asphalt mixture using X-ray CT and fractal theory, and they quantitatively analyzed the relationship between the void fraction and void equivalent diameter. Jiang et al. [21] analyzed the meso void characteristics of a porous asphalt mixture using X-ray CT, image processing, and reconstruction technology. They also investigated the influence of the void fraction, coarse and fine gradation, and nominal maximum particle size on the meso characteristics of porous asphalt mixture, as well as the relationship between meso void characteristics and mixture performance.

As a typical porous material, irregular reflection, refraction, diffraction, and other physical phenomena [22] occur in the porous asphalt pavement structure during the propagation of sound waves. Sound energy is continuously lost in the propagation process owing to viscosity and the heat conduction effect [23].This effectively reduces the traffic noise generated by pneumatic compression blasting when tires contact the pavement [24]. Several studies have investigated this phenomenon using the reverberation chamber method, standing wave ratio method, and other indoor and field tests. Losa et al. [25] developed an experimental model to predict the rolling noise of a reference car tire as a function of the composition and volumetric characteristics of mixes obtained from in-service pavements. Gonzaloet al. [26] compared CPX and road texture measurements on rubberized and standard road surfaces and studied the interaction between texture and tire/road noise. Loon et al. [27] gathered data from several countries in Europe on the age-related performance of several types of road surfaces; they studied the mechanisms that cause the deterioration based on the spectral fingerprint of the wear processes. Licitra et al. [28]

analyzed a complex interacting system that is composed of three main elements: pavement type, traffic loads, and climatic conditions. Then, they applied a new regression model to estimate the acoustic aging of the investigated pavements. Donavan [29] used the standing wave tube method to examine the sound absorption characteristics of a conventional asphalt mixture and high viscosity rubber asphalt mixture by controlling the void fraction and air resistance conditions. The authors also conducted tests on test roads paved in the USA and compared the noise reduction performance of pavements with various asphalt mixture mixtures pavements. Chu et al. [30] used the impedance tube method to compare the effects of a porous asphalt mixture (12%, 16%, 20%, 25%) and an ordinary dense graded asphalt mixture on the sound absorption performance of the tire/pavement. Bianco et al. [31] developed a measurement instrument that can be fastened to a vehicle, which overcame the issues that afflict traditional in-situ measurements. Then, the stabilization of this device and the measurement system itself were evaluated and compared with the traditional one. Knabben et al. [32,33] prepared four different types of asphalt mixtures. The sound absorption characteristics of the materials were analyzed using the standing wave tube sound absorption coefficient method; the CPX trailer method was used to characterize the tire pavement noise of Brazilian roads, and different road types were examined. Nuria et al. [34] proposed a new method based on CPX to evaluate the total tire/road sound power level emitted by an entire set of vehicle tires.

It is evident from the above literature review that X-ray CT technology and digital image processing method shave been widely used to determine void characteristic parameters such as the void fraction and connected void fraction of asphalt mixture. Furthermore, indoor and outdoor detection methods have also been used to determine the acoustic characteristics of an asphalt mixture; however, the relationship between these characteristics has not yet been elucidated. This study makes a unique contribution to the literature by investigating the relationship between the void characteristics and sound absorption characteristics of an epoxy porous asphalt mixture. Marshall specimens under different void fractions were scanned and reconstructed using X-ray CT technology and the digital image processing method. Then, void characteristic parameters such as the void fraction, connected void fraction, and curvature were obtained. An acoustic impedance tube was used to obtain the sound absorption coefficient for different frequencies. Furthermore, the relationship between these characteristics provides the technical framework for research into the noise reduction performance of porous asphalt pavement.

## 2. Materials and Methods

### 2.1. Materials

In this study, epoxy porous asphalt mixture EPA-13 was used to form Marshall specimens with different voids. The gradation is presented in Table 1. The asphalt contents were 5.5%, 5.1%, 4.7%, and 4.3%. The measured volume parameters of the Marshall specimens are listed in Table 2. The void fraction and connected void fraction were calculated according to the T-0708 volume method in the specifications for JTG E20–2011 [35]. The conventional road performance [36,37] of different asphalt mixtures is listed in Table 3.

**Table 1.** EPA-13 mineral aggregate gradation.

| Asphalt Content (%) | Passing Percentage (%) | | | | | | | | | |
|---|---|---|---|---|---|---|---|---|---|---|
| | 16 mm | 13.2 mm | 9.5 mm | 4.75 mm | 2.36 mm | 1.18 mm | 0.6 mm | 0.3 mm | 0.15 mm | 0.075 mm |
| 5.5 | 100.0 | 96.6 | 66.5 | 18.9 | 17.5 | 12.2 | 9.7 | 7.1 | 5.8 | 5.0 |
| 5.1 | 100.0 | 96.5 | 65.1 | 17.4 | 16.0 | 11.2 | 8.9 | 6.6 | 5.4 | 4.7 |
| 4.7 | 100.0 | 96.3 | 63.2 | 15.4 | 14.0 | 9.9 | 7.9 | 6.0 | 4.9 | 4.3 |
| 4.3 | 100.0 | 96.1 | 61.3 | 13.4 | 12.0 | 8.6 | 6.9 | 5.3 | 4.5 | 3.9 |

**Table 2.** EPA-13 measured volume parameters of the Marshall test.

| Asphalt Content (%) | Average Height (mm) | Specimen Volume (cm³) | Void Fraction (%) | Connected Void Fraction (%) |
|---|---|---|---|---|
| 5.5 | 62.20 | 504.28 | 17.24 | 11.19 |
| 5.1 | 63.00 | 510.76 | 18.77 | 13.60 |
| 4.7 | 63.50 | 514.81 | 20.38 | 15.76 |
| 4.3 | 63.53 | 515.06 | 21.81 | 18.21 |

**Table 3.** EPA-13 road performance.

| Asphalt Content (%) | Marshall Residual Stability (%) | Freeze–Thaw Splitting Tensile Strength Ratio (%) | Mass Loss After Cantabro Test (%) | Permeability Coefficient (ml/min) |
|---|---|---|---|---|
| 5.5 | 98.8 | 96.2 | 4.7 | 3120 |
| 5.1 | 98.6 | 95.4 | 5.1 | 3512 |
| 4.7 | 97.1 | 93.5 | 6.2 | 3943 |
| 4.3 | 96.2 | 92.1 | 7.1 | 4335 |

*2.2. Test Methods*

2.2.1. X-ray CT Scanning Void Characteristics

After obtaining the tomography image of the asphalt mixture specimen using the X-ray CT scanning system, the gray value of the image was defined based on the density of the material. The difference between the gray values of the void area and the solid area was significant; therefore, the Otsu method was used to determine the gray threshold between the void, aggregate, and asphalt mortar to extract the void part. Then, three-dimensional reconstructions were carried out after preliminary processing of the original two-dimensional image, as shown in Figure 1. The selection of an appropriate threshold is the key to processing CT scanned images into black and white binary images. The specific operation process is described below. The void fraction of each layer and the overall average void fraction were calculated using a particular gray threshold. Then, the overall average void fraction was compared with the indoor measured void fraction, and the threshold trial calculation was continuously adjusted until the total difference between the two was within a reasonable range. Finally, the characteristic parameters of the void structure of the asphalt mixture were calculated based on the determined gray threshold. X-ray CT digital analysis equipment was used for the test. This equipment is able to complete the scanning of the static physical parameters of samples and visually represent the characteristics of samples, such as the void fraction distribution, density change, crack, and cavity.

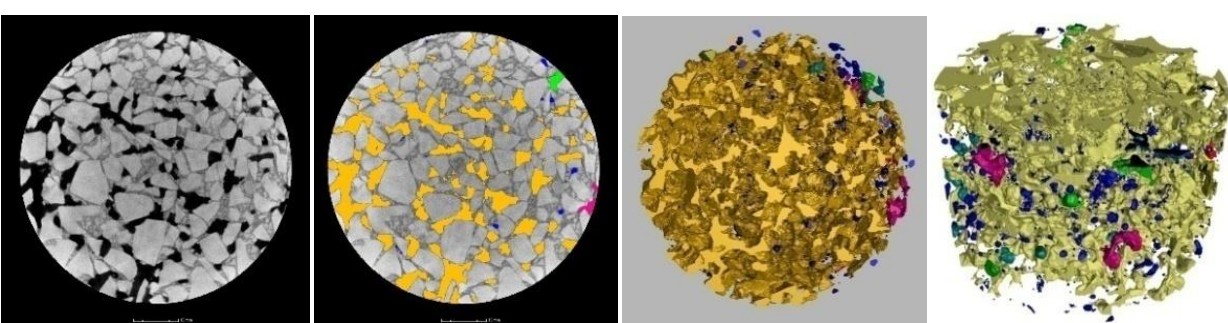

**Figure 1.** Three-dimensional reconstruction process of CT scan.

(1)    Void Fraction

Void fraction refers to the ratio of the void volume to the total volume of the sample. The ratio of the void area to the total area in a single CT image is known as the surface void

fraction. The ratio of the total void to the total volume in each section in the CT images of the specimen is known as the bulk void fraction. These ratios are expressed as follows:

$$n_m = \frac{A_V}{A_T} \tag{1}$$

$$n_t = \left[\sum_{i=1}^{N} \frac{(A_V)_i}{A_T}\right] N^{-1} \tag{2}$$

where $n_m$ is the surface void fraction, $n_t$ is the bulk void fraction, $N$ is the total number of images, $A_V$ is the void area of a single image, and $A_T$ is the total area of a single image.

(2)    Connected Void Fraction

Void connectivity is an important three-dimensional void characteristic of asphalt mixture and a channel for water migration and drainage. When analyzing vertically (Z-axis) connected voids, it is necessary to track all connected voids from the top to the bottom of the specimen and eliminate all disconnected voids owing to the anisotropy of voids in the asphalt mixture; furthermore, the connected voids in different coordinate directions vary. The specific identification process of vertically (Z-axis) connected voids is assumed a value of 0 for a void. Each binary image is identified, and when the value is 0, the nearest eight values around it are checked. If one of the eight values is 0, all the values in the area with 0 are classified into one category. The void is considered to be connected in the area and is marked as the same void; the same steps are repeated until the last image has been checked. Finally, the images are individually checked from the first image at the top to the last image at the bottom, and all connected structures are defined as connected voids. The recognition process of horizontally connected voids is similar.

(3)    Curvature

The drainage channels in the three-dimensional connected void space of an asphalt mixture are very complex, and there are many cross-path types. It is difficult to calculate the curvature of each drainage channel individually. Therefore, the independent void sections in each image slice are set and grouped after obtaining the connected void structures in the X, Y, and Z directions. The centroid of each group of voids is connected to the centroid of the nearest void section in the image by a straight line. The ratio of the length of the connecting line (i.e., the actual flow length $L_e$) to the vertical distance between the two sectional images (i.e., the vertical length L of the inlet and outlet of the flow path) is defined as the curvature $T$, as shown in Figure 2, and the corresponding equation is given in Formula (3). The curvature of the entire specimen is equal to the average value of the curvatures of all continuous image slices.

$$T = \frac{L_e}{L} \tag{3}$$

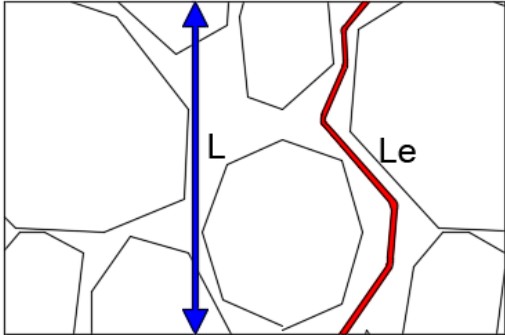

**Figure 2.** Schematic of a curvature.

### 2.2.2. Sound Absorption Characteristics

In this study, an acoustic impedance tube was used to detect the sound absorption coefficient of a porous asphalt mixture. An acoustic impedance tube system consists of an impedance tube, high-end microphone, data acquisition card, computer, etc. The pipe diameter is 100 mm, and the measurement frequency range is 63~1600 Hz. The incident sound wave $P_i$ is generated using a loudspeaker at one end of the acoustic impedance tube. Then, $P_i$ is superimposed with the sound wave $P_r$ that is reflected from the surface of the test piece to form a standing wave in the acoustic impedance tube, as shown in Figure 3. The standing wave ratio is obtained based on the sound pressure amplitude $|P(x_{\max,n})|$ at the maximum sound pressure and $|P(x_{\min,n})|$ at the minimum sound pressure. The sound absorption coefficient of the porous asphalt mixture can be obtained through calculation.

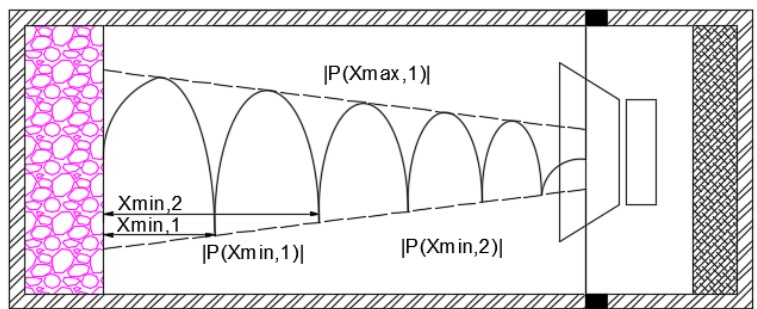

**Figure 3.** Schematic of acoustic impedance tube detection.

When $P_i(x)$ and $P_r(x)$ are in phase, a maximum occurs on the standing wave diagram. That is,

$$|P_{\max}| = |P_0| \cdot (1 + |r|) \tag{4}$$

When $P_i(x)$ and $P_r(x)$ are completely out of phase, a minimum occurs. That is,

$$|P_{\min}| = |P_0| \cdot (1 - |r|) \tag{5}$$

In the above equations, $P_{\max}$ is the maximum value of the sound pressure, $P_{\min}$ is the minimum value of the sound pressure, $P_0$ is the sound pressure amplitude, and $r$ is the reflection factor.

The sound absorption coefficient $\alpha$ is given as

$$\alpha = \frac{4 \times 10^{\frac{\Delta L}{20}}}{10^{\frac{\Delta L}{20}} + 1} \tag{6}$$

where $\Delta L$ is the difference in the sound pressure level between the maximum sound pressure $|P_{\max}|$ and minimum sound pressure $|P_{\min}|$.

## 3. Results

### 3.1. Analysis of CT Scanning Void Characteristics

In this test, the vertical scanning interval of the epoxy porous asphalt mixture specimens is 0.105 mm, and the radial scanning interval is 2°. To reduce the analysis error, a 3 mm thick layer is removed from the upper and lower surfaces, and a 9 mm thick layer is also removed in the diameter direction based on the image quality, as shown in Figure 4. The overall void characteristic parameters in the X-ray CT scanning area are presented in Table 4.

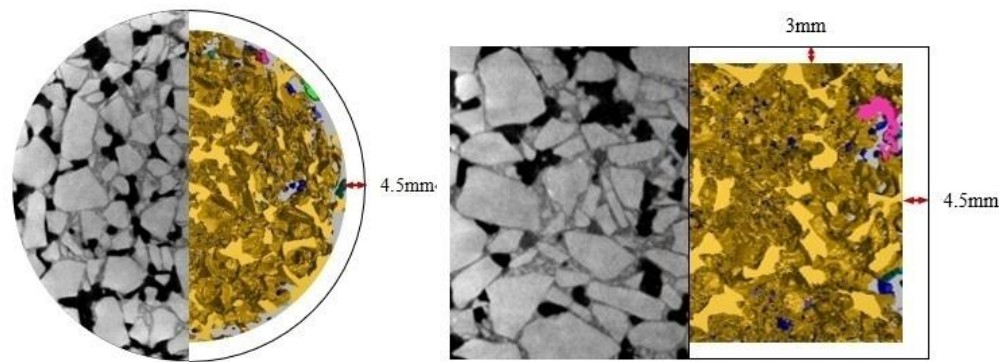

**Figure 4.** EPA-13 scanning calculation area.

**Table 4.** Characteristic parameters of EPA-13 CT scanning void.

| Asphalt Content (%) | Calculation Area Volume (cm³) | Calculation Area Void Volume (cm³) | Calculation Area Void Fraction (%) | Calculation Area Connected Void Fraction (%) |
| --- | --- | --- | --- | --- |
| 5.5 | 378.48 | 50.11 | 13.24 | 11.72 |
| 5.1 | 383.87 | 55.77 | 14.53 | 13.12 |
| 4.7 | 387.24 | 61.25 | 15.82 | 14.44 |
| 4.3 | 387.44 | 67.07 | 17.31 | 16.03 |

Figure 5 shows a comparison of the measured void characteristic parameters of the Marshall specimen (Table 2) and the CT scanning void characteristic parameters (Table 4).

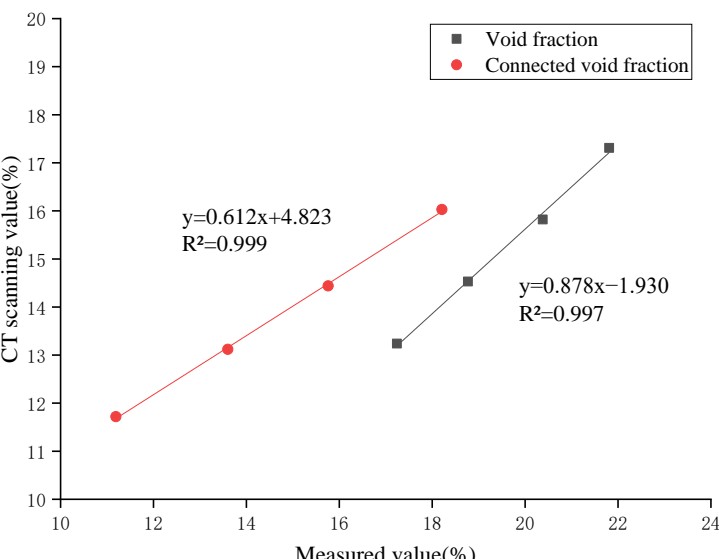

**Figure 5.** Comparison of the measured and CT scanning values of the void characteristic parameters of EPA-13.

It can be observed from Figure 5 that there is a good correlation (higher than 0.99) between the void characteristics of the epoxy porous asphalt mixture obtained using CT scanning (mesoscale) and the measured values (macroscale). The measured values are generally higher than the CT scanning values, primarily owing to the large macrostructure of the outer layer of the specimen. When the volume method is used for the calculation, the depth of the structure is considered the void, thereby resulting in a higher measured value. In addition, the difference between the void fraction and connected void fraction gradually decreases with an increase in the void fraction, which indicates that the proportion of

the connected void fraction in the void fraction further increases after the void structure reaches a particular degree.

### 3.1.1. Surface Void Analysis

(1)　Vertical Scanning

The image voids of each layer along the height direction of the specimen are calculated based on the vertical CT scanning sectional images of different specimens, as presented in Table 5, and the distribution characteristics of voids in the vertical position are shown in Figure 6.

**Table 5.** EPA-13 surface voids at different vertical section positions.

| Asphalt Content 5.5% | | Asphalt Content 5.1% | | Asphalt Content 4.7% | | Asphalt Content 4.3% | |
|---|---|---|---|---|---|---|---|
| Section Position (mm) | Surface Void Fraction (%) | Section Position (mm) | Surface Void Fraction (%) | Section Position (mm) | Surface Void Fraction (%) | Section Position (mm) | Surface Void Fraction (%) |
| 0.105 | 20.339 | 0.105 | 23.12 | 0.105 | 26.455 | 0.105 | 29.108 |
| 0.210 | 19.765 | 0.210 | 22.577 | 0.210 | 25.488 | 0.210 | 29.026 |
| 0.315 | 19.133 | 0.315 | 22.101 | 0.315 | 24.631 | 0.315 | 28.296 |
| 0.420 | 18.541 | 0.420 | 21.731 | 0.420 | 23.758 | 0.420 | 28.087 |
| . . . | . . . | . . . | . . . | . . . | . . . | . . . | . . . |
| 55.755 | 26.345 | 56.595 | 29.692 | 57.120 | 31.874 | 57.120 | 32.222 |
| 55.860 | 27.096 | 56.700 | 30.595 | 57.225 | 32.461 | 57.225 | 32.996 |
| 55.965 | 28.031 | 56.805 | 31.496 | 57.330 | 33.129 | 57.330 | 33.872 |
| 56.070 | 28.946 | 56.910 | 32.432 | 57.435 | 33.745 | 57.435 | 34.747 |

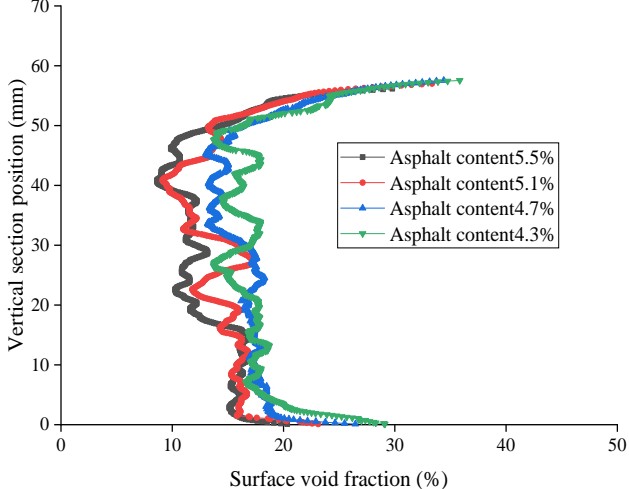

**Figure 6.** Surface void fraction distribution trend of EPA-13 vertical section position.

It can be observed from Figure 6 that the voids at both ends of the EPA-13 specimens are large, and the formed upper surface is larger than the lower surface. The voids near the lower surface are relatively uniform, whereas the voids near the upper surface fluctuate significantly. This phenomenon is primarily owing to the forming method. Compaction mainly depends on gravity being transmitted from top to bottom. The coarse aggregate does not move easily owing to the skeleton, whereas the fine aggregate and mortar still move under the action of an external force. In the second impact, most of the mortar is embedded in the void by the coarse aggregate owing to the stability of the structure, and the final lower area that is formed is more uniform than the upper area.

(2)　Radial Scanning

The image voids of each layer along the diameter direction of the specimen are calculated based on the radial CT scanning sectional images of different specimens, as presented in Table 6. The distribution characteristics of the voids in the radial position are shown in Figure 7.

**Table 6.** EPA-13 surface voids at different radial section positions.

| Section Position Angle (°) | Surface Void Fraction (%) | | | |
| --- | --- | --- | --- | --- |
| | Asphalt Content 5.5% | Asphalt Content 5.1% | Asphalt Content 4.7% | Asphalt Content 4.3% |
| 2 | 21.636 | 23.463 | 25.929 | 28.289 |
| 4 | 20.610 | 22.409 | 23.702 | 26.600 |
| 6 | 19.403 | 20.923 | 21.912 | 23.619 |
| 8 | 18.687 | 19.455 | 19.768 | 22.219 |
| . . . | . . . | . . . | . . . | . . . |
| 354 | 19.651 | 20.031 | 20.779 | 24.635 |
| 356 | 20.933 | 20.118 | 22.461 | 25.652 |
| 358 | 21.176 | 21.426 | 24.168 | 26.255 |
| 360 | 21.031 | 23.586 | 25.245 | 28.301 |

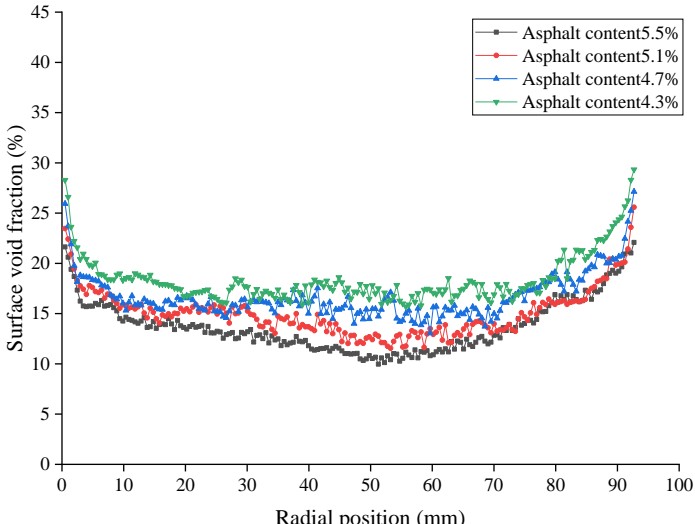

**Figure 7.** Surface void fraction distribution trend of EPA-13 radial section position.

It can be observed from Figure 7 that the outer void fractions of the EPA-13 specimen are slightly larger than the inner values. The void fraction gradually decreases as the section approaches the axis, and the radial void fraction gradually increases with a decrease in asphalt content. The distribution of the void fraction on a radial surface is more uniform compared with that on a vertical scanning surface.

### 3.1.2. Bulk Void Analysis

The epoxy porous asphalt mixture Marshall specimen is reconstructed in three dimensions based on the CT scanning sectional image, and the volume parameters of each void are extracted. Finally, the void distribution characteristics of different specimens are obtained, as listed in Tables 7–10. Bulk void 1 is the connected void.

The number of bulk voids in the EPA-13 specimens in the intervals 0.01–0.1 mm$^3$, 0.1–1 mm$^3$, 1–10 mm$^3$, 10–20 mm$^3$, 20–40 mm$^3$, 40–60 mm$^3$, 60–80 mm$^3$, 80–100 mm$^3$, 100–200 mm$^3$, and 200–100,000 mm$^3$ is counted based on the extracted bulk void parameters under different asphalt contents. The percentage of voids in the total number from all specimens is presented in Table 11 and Figure 8.

**Table 7.** Characteristic parameters of bulk voids in EPA-13 with 5.5% asphalt content.

| Bulk Void | Void Volume (mm$^3$) | Equivalent Diameter (mm) | Surface Area (mm$^2$) |
|---|---|---|---|
| 1 | 44361.40 | 43.92 | 134396.76 |
| 2 | 153.94 | 6.65 | 419.60 |
| 3 | 142.28 | 6.48 | 373.73 |
| 4 | 87.79 | 5.51 | 229.90 |
| 5 | 83.53 | 5.42 | 278.04 |
| . . . | . . . | . . . | . . . |
| 2394 | 0.01 | 0.27 | 0.37 |
| 2395 | 0.01 | 0.27 | 0.35 |
| 2396 | 0.01 | 0.27 | 0.31 |
| 2397 | 0.01 | 0.27 | 0.26 |
| 2398 | 0.01 | 0.27 | 0.37 |

**Table 8.** Characteristic parameters of bulk voids in EPA-13 with 5.1% asphalt content.

| Bulk Void | Void Volume (mm$^3$) | Equivalent Diameter (mm) | Surface Area (mm$^2$) |
|---|---|---|---|
| 1 | 50375.43 | 45.82 | 137436.09 |
| 2 | 367.72 | 8.89 | 934.31 |
| 3 | 177.72 | 6.98 | 540.78 |
| 4 | 129.32 | 6.27 | 368.74 |
| 5 | 94.12 | 5.64 | 290.17 |
| . . . | . . . | . . . | . . . |
| 1825 | 0.01 | 0.27 | 0.35 |
| 1826 | 0.01 | 0.27 | 0.33 |
| 1827 | 0.01 | 0.27 | 0.31 |
| 1828 | 0.01 | 0.27 | 0.26 |
| 1829 | 0.01 | 0.27 | 0.33 |

**Table 9.** Characteristic parameters of bulk voids in EPA-13 with 4.7% asphalt content.

| Bulk Void | Void Volume (mm$^3$) | Equivalent Diameter (mm) | Surface Area (mm$^2$) |
|---|---|---|---|
| 1 | 55918.66 | 47.44 | 139672.11 |
| 2 | 210.19 | 7.38 | 624.29 |
| 3 | 178.61 | 6.99 | 439.11 |
| 4 | 173.56 | 6.92 | 585.76 |
| 5 | 89.41 | 5.55 | 267.76 |
| . . . | . . . | . . . | . . . |
| 1660 | 0.01 | 0.27 | 0.33 |
| 1661 | 0.01 | 0.27 | 0.33 |
| 1662 | 0.01 | 0.27 | 0.35 |
| 1663 | 0.01 | 0.27 | 0.35 |
| 1664 | 0.01 | 0.27 | 0.35 |

**Table 10.** Characteristic parameters of bulk voids in EPA-13 with 4.3% asphalt content.

| Bulk Void | Void Volume (mm$^3$) | Equivalent Diameter (mm) | Surface Area (mm$^2$) |
|---|---|---|---|
| 1 | 62095.26 | 49.13 | 145212.58 |
| 2 | 136.86 | 6.39 | 437.26 |
| 3 | 84.05 | 5.43 | 208.05 |
| 4 | 47.91 | 4.51 | 183.65 |
| 5 | 31.14 | 3.90 | 92.38 |
| . . . | . . . | . . . | . . . |
| 2289 | 0.01 | 0.27 | 0.33 |
| 2290 | 0.01 | 0.27 | 0.35 |
| 2291 | 0.01 | 0.27 | 0.33 |
| 2292 | 0.01 | 0.27 | 0.37 |
| 2293 | 0.01 | 0.27 | 0.33 |

**Table 11.** Bulk void quantity distribution of EPA-13 under different asphalt contents.

| Volume Interval (mm³) | Asphalt Content 5.5% | Asphalt Content 5.1% | Asphalt Content 4.7% | Asphalt Content 4.3% |
|---|---|---|---|---|
| 0.01–0.1 | 1311 | 1272 | 997 | 1661 |
| 0.1–1 | 828 | 392 | 520 | 522 |
| 1–10 | 196 | 129 | 106 | 88 |
| 10–20 | 28 | 15 | 25 | 11 |
| 20–40 | 21 | 11 | 7 | 7 |
| 40–60 | 6 | 3 | 2 | 1 |
| 60–80 | 3 | 2 | 2 | 0 |
| 80–100 | 2 | 1 | 1 | 1 |
| 100–200 | 2 | 2 | 2 | 1 |
| 200–100,000 | 1 | 2 | 2 | 1 |

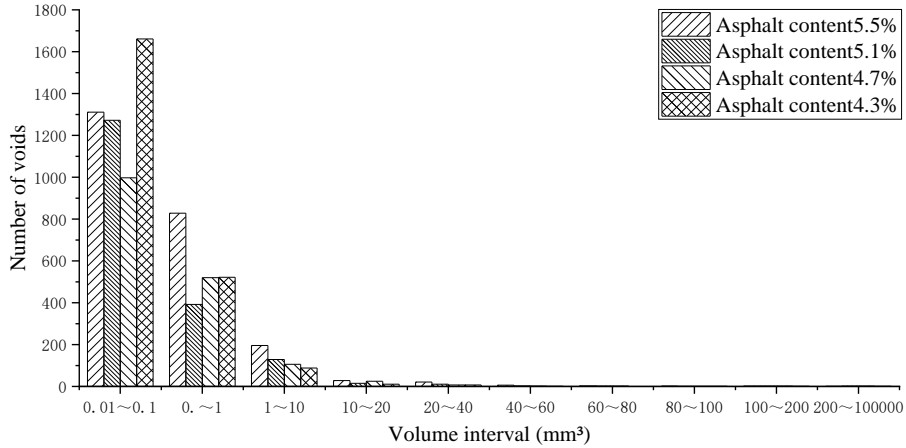

**Figure 8.** Bulk void quantity distribution of EPA-13 under different asphalt contents.

According to the test results, the number of voids in the porous asphalt mixture specimen is very high, and the closed voids below 1 mm³ account for more than 89% of all voids.

### 3.1.3. Connected Voids Analysis

The connected void volume, connected void fraction, connected void equivalent diameter, connected void surface area, and other parameters are calculated using CT scanning data. The calculation results are presented in Table 12.

**Table 12.** Characteristic parameters of the connected void in EPA-13 using CT scanning.

| Asphalt Content (%) | Void Fraction (%) | Connected Void Fraction (%) | Connected Void Volume (cm³) | Equivalent Diameter (mm) | Surface Area (mm²) | Curvature |
|---|---|---|---|---|---|---|
| 5.5 | 13.24 | 11.72 | 44.36 | 43.92 | 134,396.76 | 2.573 |
| 5.1 | 14.53 | 11.12 | 50.38 | 45.82 | 137,436.09 | 2.439 |
| 4.7 | 15.82 | 14.44 | 55.92 | 47.44 | 139,672.11 | 2.273 |
| 4.3 | 17.31 | 16.03 | 62.10 | 49.13 | 145,212.58 | 2.077 |

It can be observed from the test results that the connected voids gradually increase with a decrease in the asphalt content. The $R^2$ values of the void fraction and connected void fraction are up to 0.999 (Figure 9). Therefore, the connected void fraction can be predicted when the premise of the void fraction is known.

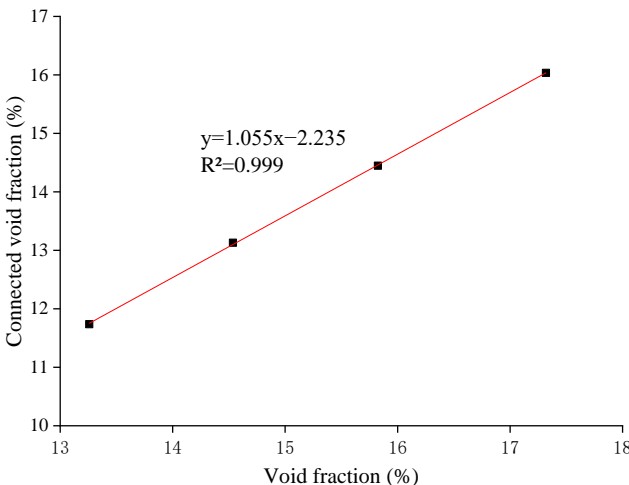

**Figure 9.** EPA-13 void fraction and connected void fraction fitting curve.

### 3.1.4. Curvature Analysis

The curvature calculation results of the connected void of EPA-13 are listed in Table 12 for different void fractions.

It can be observed from the test results that a larger connected void fraction corresponds to a smaller curvature. The correlation coefficient between the connected void fraction and curvature is 0.997 (Figure 10), and the correlation coefficient between the surface area and curvature is 0.981, indicating a good correlation.

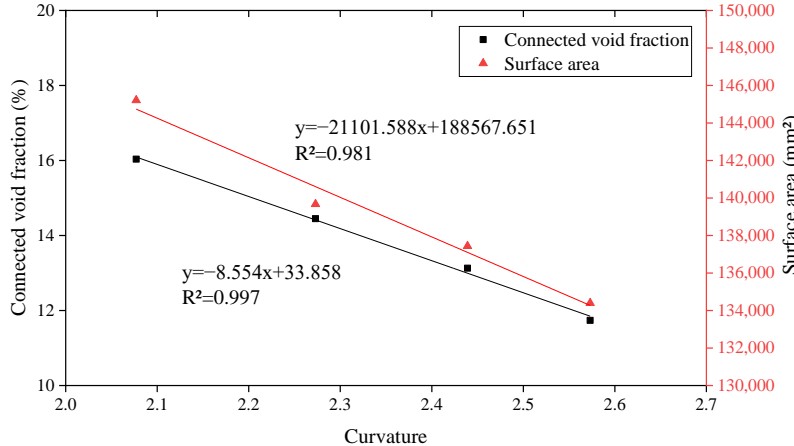

**Figure 10.** Relationship curve between the connected void fraction, surface area, and curvature curve for EPA-13.

### 3.2. Sound Absorption Characteristics of Acoustic Impedance Tube Analysis

The sound absorption coefficient test results of the epoxy porous asphalt mixture specimen at different frequencies are shown in Figure 11.

According to the test results, the relationship curve between the sound absorption coefficient and frequency exhibits a double peak trend. The peak value of the sound absorption coefficient increases with an increase in the void fraction, and the trend is clear at the first peak. The peak sound absorption coefficient of the EPA-13 specimen with an asphalt content of 5.5% is 0.517, and the first peak frequency is approximately 440 Hz. The peak sound absorption coefficient of the EPA-13 specimen with an asphalt content of 5.1% is 0.676, and the first peak frequency is approximately 500 Hz. The peak sound absorption coefficient of the EPA-13 specimen with an asphalt content of 4.7% is 0.807, and the first peak frequency is approximately 540 Hz. The peak sound absorption coefficient of

the EPA-13 specimen with an asphalt content of 4.3% is 0.937, and the first peak frequency is approximately 560 Hz. It can be observed that an increase in the void fraction shifts the frequency corresponding to the peak of $\alpha$ in the high-frequency direction, which indicates that an increase in voidage is conducive to the absorption of high-frequency noise by the asphalt mixture.

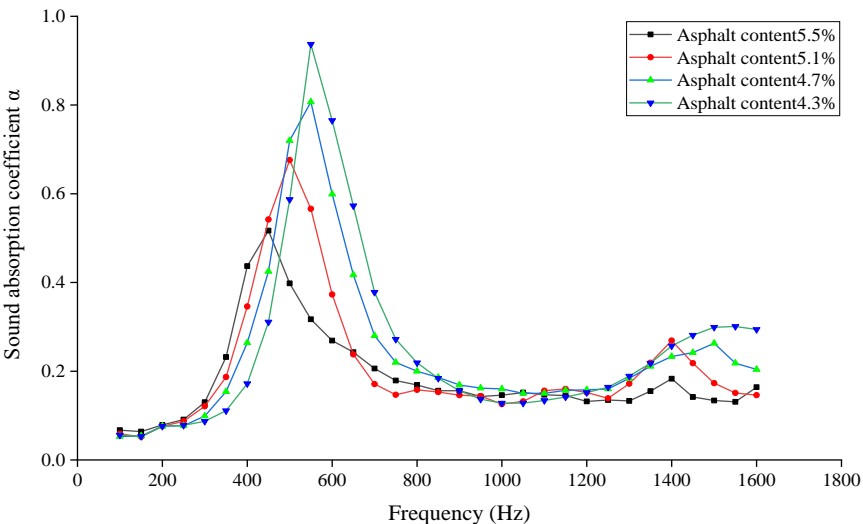

**Figure 11.** Variation trend of EPA-13 sound absorption coefficient with different asphalt contents.

## 4. Discussion

Gray theory was used to analyze the correlation between void characteristic parameters and sound absorption characteristics. Gray theory is a systematic analysis technology [38] that performs a quantitative comparative analysis of a development trend according to its similarity or difference between factors. It also determines the main factors that affect the target value by evaluating the correlation between the target value (reference sequence) and the influencing factors (comparison sequence). By taking the peak and average values of the sound absorption coefficient as the reference series, the void characteristic parameters such as the void fraction, connected void fraction, equivalent diameter, surface area, and curvature can be calculated as the comparison series. The calculation results are presented in Tables 13 and 14.

According to the calculation results, the order of the correlation degree between the peak and average values of the sound absorption coefficient and the characteristic parameters of the voids is: connected void fraction > void fraction > equivalent diameter of connected void fraction > surface area of connected void fraction > curvature. This indicates that the connected void fraction has the greatest effect on the sound absorption coefficient.

The peak and average values of the sound absorption coefficient were fitted to the connected void fraction, and the fitting curve is shown in Figure 12.

**Table 13.** Gray correlation analysis of the peak sound absorption coefficient and void characteristic parameters.

| Factor | | Asphalt Content (%) | | | | Correlation Degree R |
|---|---|---|---|---|---|---|
| | | **5.5** | **5.1** | **4.7** | **4.3** | |
| Reference sequence | $\alpha$ peak value | 0.517 | 0.676 | 0.807 | 0.937 | |
| Comparison sequence | Void fraction (%) | 13.24 | 14.53 | 15.82 | 17.31 | 0.6953 |
| | Connected void fraction (%) | 11.72 | 13.12 | 14.44 | 16.03 | 0.7153 |
| | Equivalent diameter (mm) | 43.92 | 45.82 | 47.44 | 49.13 | 0.6463 |
| | Surface area (mm$^2$) | 134,396.76 | 137,436.09 | 139,672.11 | 145,212.58 | 0.6336 |
| | Curvature | 2.57 | 2.44 | 2.27 | 2.08 | 0.5855 |

**Table 14.** Gray correlation analysis of the average sound absorption coefficient and void characteristic parameters.

| Factor | | Asphalt Content (%) | | | | Correlation Degree R |
|---|---|---|---|---|---|---|
| | | 5.5 | 5.1 | 4.7 | 4.3 | |
| Reference sequence | $\alpha$ average value | 0.186 | 0.208 | 0.240 | 0.253 | |
| Comparison sequence | Void fraction (%) | 13.24 | 14.53 | 15.82 | 17.31 | 0.8778 |
| | Connected void fraction (%) | 11.72 | 13.12 | 14.44 | 16.03 | 0.9486 |
| | Equivalent diameter (mm) | 43.92 | 45.82 | 47.44 | 49.13 | 0.7215 |
| | Surface area (mm$^2$) | 134,396.76 | 137,436.09 | 139,672.11 | 145,212.58 | 0.6904 |
| | Curvature | 2.57 | 2.44 | 2.27 | 2.08 | 0.5892 |

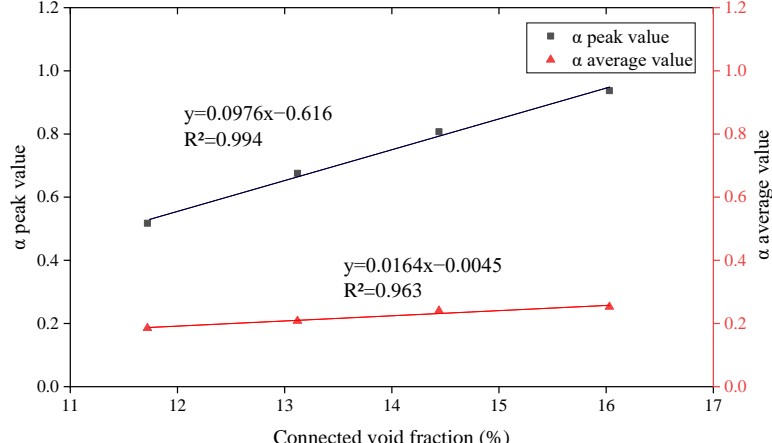

**Figure 12.** Fitting curve of the peak and average values of the sound absorption coefficient with the connected void fraction.

As shown in Figure 12, the correlation coefficient $R^2$ between the connected void fraction and the peak value of the sound absorption coefficient is 0.994. The correlation coefficient $R^2$; between the connected void fraction and the average value of the sound absorption coefficient is 0.963, which indicates that it has a high correlation. Therefore, the sound absorption coefficient can be predicted and analyzed based on the connected void fraction.

## 5. Conclusions

In this study, the void characteristics of an epoxy porous asphalt mixture obtained using CT scanning (mesoscale) exhibited a good correlation with the measured values (macroscale). This indicated that the combination of CT scanning technology and digital image processing could not only characterize the complex meso morphological characteristics and spatial distribution of voids in the asphalt mixture but also accurately characterize the overall size of the voids. This provides a good entry point to better analyze the void characteristics of the asphalt mixture. The order of the gray correlation degree between the peak and average values of the sound absorption coefficient and the void characteristic parameters is as follows: connected void fraction > void fraction > equivalent diameter of connected void > surface area of connected void > curvature, indicating the sound absorption coefficient can be predicted and analyzed based on the connected void fraction. In the next step, this technology can be used to study the drainage characteristics and road surface skid resistance of porous pavement.

**Author Contributions:** X.L. and T.L. conceived and designed the experiments; H.D., J.J., and X.Z. conducted the experiments; J.G. and H.D. analyzed the data; X.L. wrote the paper. All authors have read and agreed to the published version of the manuscript.

**Funding:** This research received no external funding.

**Institutional Review Board Statement:** Not applicable.

**Informed Consent Statement:** Not applicable.

**Data Availability Statement:** Not applicable.

**Conflicts of Interest:** The authors declare no conflict of interest regarding the publication of this paper.

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
