# Peer review of "Relationship between the Void and Sound Absorption Characteristics of Epoxy Porous Asphalt Mixture Based on CT"

_coatings, doi:10.3390/coatings12030328_

Round 1

Reviewer 1 Report

  • The abstract should be improved. Especially, the flow of the language. I should start with a short introduction, objectives, and major conclusions.
  • Your introduction needs more information about performance. I missing some general information regarding the performance of asphalt mixtures such as https://doi.org/10.1016/j.conbuildmat.2017.07.164 and doi.org/10.1080/14680629.2021.1908408.
  • Please state clearly your objectives.
  • Please provide mix design data.
  • Your conclusions are too long. Don`t repeat your disscution.

Author Response

Point 1: The abstract should be improved. Especially, the flow of the language. It should start with a short introduction, objectives, and major conclusions.

Response 1: This study investigates the relationship between the void characteristics and sound absorption characteristics of epoxy porous asphalt mixture. The specimens are scanned and reconstructed under different void fractions using X-ray computed tomography (CT) technology and digital image processing, and the sound absorption coefficients at different frequencies are obtained using an acoustic impedance tube. The relationship between void characteristics and sound absorption characteristics is analyzed using grey correlation. The test results exhibited a good correlation between the void characteristics of epoxy porous asphalt mixture obtained by CT scanning (meso scale) and the measured values (macro scale). The difference between the void fraction and connected void fraction gradually decreased with an increase in the void fraction. The relationship curve between the sound absorption coefficient and frequency exhibited a bimodal trend, and the peak value of the sound absorption coefficient increased with an increase in the void fraction. The order of the grey correlation degree between the peak and average values of the sound absorption coefficient and the void characteristic parameters is as follows: connected void fraction > void fraction > equivalent diameter of connected void > surface area of connected void > curvature.”

Point 2: Your introduction needs more information about performance. I missing some general information regarding the performance of asphalt mixtures such as https://doi.org/10.1016/j.conbuildmat.2017.07.164 and doi.org/10.1080/14680629.2021.1908408.

Response 2: Thanks for your careful review and comments. The conventional road performance [36,37] of different asphalt mixtures are listed in Table 3.

Table 3. EPA -13 road performance.

Asphalt content (%)

Marshall residual stability (%)

Freeze-thaw splitting tensile strengthratio (%)

Mass loss after Cantabro test (%)

Permeability coefficient (ml/min)

5.5

98.8

96.2

4.7

3,120

5.1

98.6

95.4

5.1

3,512

4.7

97.1

93.5

6.2

3,943

4.3

96.2

92.1

7.1

4,335

Point 3: Please state clearly your objectives.

Response 3:  Thanks for your careful review and comments.It is clear from the above literature review that X-ray CT technology and digital image processing method were used to determine void characteristic parameters, meanwhile, indoor and outdoor detection methods were also used to determine the acoustic characteristics of asphalt mixture; however, the relationship between them has not yet been elucidated. In order to study the relationship between void characteristics and sound absorption characteristics of epoxy porous asphalt mixture, this paper used the X-ray CT technology and digital image processing method to obtain the void characteristic parameters of the specimen, the sound absorption coefficient was obtained by acoustic impedance tube, and then the relationship between them was studied, thus it could provide technical support for the research of noise reduction performance of the porous asphalt pavement.

Point 4: Please provide mix design data.

Response 4: Thanks for your careful review and comments. It has been improved, the synthetic gradation of mineral aggregate, asphalt content, Marshall volume index and conventional road performance are shown in this paper.

Point 5: Your conclusions are too long. Don`t repeat your disscution.

Response 5: Thanks for your careful review and comments. The conclusion has been refined and simplified as follows:

“In this study, the void characteristics and sound absorption characteristic parameters of epoxy porous asphalt mixture were tested using X-ray CT technology and an acoustic impedance tube. Their relationship was examined, and the following conclusions are drawn based on the findings of the study.

  1. The void characteristics of epoxy porous asphalt mixture obtained using CT scanning (meso scale) exhibited a good correlation with the measured values (macro scale). This indicated that the combination of CT scanning technology and digital image processing can not only characterize the complex meso morphological characteristics and spatial distribution of voids in asphalt mixture, but also accurately characterize the overall size of voids. This provides a good entry point to better analyze the void characteristics of the asphalt mixture.
  2. Both vertical and radial scanning demonstrate that the surface void fraction of the specimen is greater than the internal void fraction. The voids at the upper and lower parts are large, those in the middle part near the lower surface are relatively uniform, and those near the upper surface fluctuate significantly. The void fraction gradually decreases as the section approaches the axis. In addition, the number of voids in the EPA-13 epoxy porous asphalt mixture specimen is very high, and the closed voids below 1 mm³ account for more than 89% of all voids.
  3. The order of the grey correlation degree between the peak and average values of the sound absorption coefficient and the void characteristic parameters is as follows: connected void fraction > void fraction > equivalent diameter of connected void > surface area of connected void > curvature. Moreover, the correlation coefficient R² is very high. Therefore, the sound absorption coefficient can be predicted and analyzed based on the connected void fraction.
  4. In this study, the technical route was clear and feasible. In the next step, this technology can be used to study the drainage characteristics and road surface skid resistance of porous pavement.”

In addition, the English language and style of this paper are fully corrected and checked by native English speakers.

Reviewer 2 Report

The paper analyses the relationship between void characteristics and sound absorption of Porous Asphalt Mixes based on CT Technology. Four mixes are compared in terms of different characteristics such as connected void fraction, void fraction and curvature obtained using CT scanning and traditional methods.

The approach and proposed methods are interesting and would be worthy of publication, despite the necessity of deep revisions, following the suggestions I report below:

  • A general English style and form revision is suggested.

  • Introduction: The introduction should be improved. In order to increase paper’s quality, I would suggest to write a few lines regarding the issues related to the volumetric parameter measurements in hot mix asphalts (Alvarez, AE et al. “Internal structure of compacted permeable friction course mixtures”. Constr Build Mater 2010, 24:1027–35; Praticò, Filippo G. et al “Measurement of air void content in hot mix asphalts: Method and core diameter dependence”, Constructions and building materials 26 (2012) 344–349) and the influence of these characteristics on the phenomenon of water flows along an asphalt pavement (Murali Krishnan et al. “Permeability and bleeding of asphalt concrete using mixture theory”, International Journal of Engineering Science 39 (2001) 611–627; Praticò, Filippo G. et al “Flow of water in rigid solids: Development and experimental validation of models for tests on asphalts” Computers and Mathematics with Applications 55 (2008) 235–244)

  • Referring to tire/pavement noise generation mechanisms, the authors could introduce information about available models related to noise levels and pavement properties (Losa Massimo et al. 2013, Mixture Design Optimization of Low-Noise Pavements, Transportation Research Record: Journal of the Transportation Research Board, 2372, 25-33, (2013); De León Gonzalo et al. “Evaluation of tyre/road noise and texture interaction on rubberised and conventional pavements using CPX and profiling measurements”, Road Materials and Pavement Design, 2020, 21(S1), pp. S91-S102) and acoustic ageing phenomena (van Loon R et al. 2015. Mechanism of acoustic aging of road surfaces. In: Proceedings of EuroNoise 2015, 31 May – 3 June, Maastricht; Licitra, Gaetano et al.  “Modelling of acoustic ageing of rubberized pavements”. Applied Acoustics 146 (2019) 237–245).

  • Lines 41-44: I suggest replacing “seepage” with “drainage”.
  • Section 2.1: Please specify the reference standard used to carry out the void fraction and connected void fraction values listed in Table 2.
  • Table 2: Please insert the unit of measure of sieve size;
  • I suggest replacing “Asphalt dosage” with “Asphalt content” everywhere.
  • Line 139: The authors specify the name of the X-ray tomography CT digital analysis equipment v|tome|x s produced by General Electric Company.

I suggest to avoid product or brand names and to use general terms as “X-ray tomography CT digital analysis equipment” specifying its characteristics.

  • Lines 155-167: This section is not well explained and organised. I suggest to better explain the sequence of the identification process of connected void fraction. Why are there eight values?
  • Lines 293-294-303-304-354-356: I suggest not to use formulas within the text.
  • Tables 11-12: Table 12 is repetitive. Please add the column of curvature values in table 11 and delete table 12.

CONCLUSIONS: The conclusions are should be improved. Referring the discussion of results, the authors should emphasize the potential of the parameters obtained using CT scanning, as the possibility of predicting the sound absorption coefficient using the connected void fraction. Also, I suggest to put something about future developments at the end of the conclusions.

Author Response

Point 1: A general English style and form revision is suggested.

Response 1: Thanks for your careful review and comments. The English language style and form of this paper are fully corrected and checked by native English speakers.

Point 2: Introduction: The introduction should be improved. In order to increase paper’s quality, I would suggest to write a few lines regarding the issues related to the volumetric parameter measurements in hot mix asphalts (Alvarez, AE et al. “Internal structure of compacted permeable friction course mixtures”. Constr Build Mater 2010, 24:1027–35; Praticò, Filippo G. et al “Measurement of air void content in hot mix asphalts: Method and core diameter dependence”, Constructions and building materials 26 (2012) 344–349) and the influence of these characteristics on the phenomenon of water flows along an asphalt pavement (Murali Krishnan et al. “Permeability and bleeding of asphalt concrete using mixture theory”, International Journal of Engineering Science 39 (2001) 611–627; Praticò, Filippo G. et al “Flow of water in rigid solids: Development and experimental validation of models for tests on asphalts” Computers and Mathematics with Applications 55 (2008) 235–244)

Response 2: Thanks for your careful review and comments. The Introduction has been revised as “Evaluation indexes based on volume characteristics, such as void fraction and connected void fraction, are typically used to evaluate the void characteristics measurement and its impact on the drainage and noise reduction performance of the asphalt mixture. Alvarez et al. [3] studied the internal structure of permeable friction course mixtures and assessed it in terms of air-void characteristics. Praticò et al. [4,5] observed that both measurement methodology and core diameter can substantially affect the specific gravity determination and air-void content estimate, and they proposed the theoretical derivation of models for some tests on asphalts by referring to the general theory for the percolation of water in rigid porous materials. Krishnan et al. [6] relaxed the assumption of constant permeability in the theory of consolidation by Terzaghi, and proposed a linear relation between permeability and air voids.”

Point 3: Referring to tire/pavement noise generation mechanisms, the authors could introduce information about available models related to noise levels and pavement properties (Losa Massimo et al. 2013, Mixture Design Optimization of Low-Noise Pavements, Transportation Research Record: Journal of the Transportation Research Board, 2372, 25-33, (2013); De León Gonzalo et al. “Evaluation of tyre/road noise and texture interaction on rubberised and conventional pavements using CPX and profiling measurements”, Road Materials and Pavement Design, 2020, 21(S1), pp. S91-S102) and acoustic ageing phenomena (van Loon R et al. 2015. Mechanism of acoustic aging of road surfaces. In: Proceedings of EuroNoise 2015, 31 May – 3 June, Maastricht; Licitra, Gaetano et al.  “Modelling of acoustic ageing of rubberized pavements”. Applied Acoustics 146 (2019) 237–245).

Response 3:  Thanks for your careful review and comments. Relevant references have been added and improved. “Losa et al. [25] developed an experimental model to predict the rolling noise of a reference car tire as a function of the composition and volumetric characteristics of mixes obtained from in-service pavements. Gonzalo et al. [26] compared CPX and road texture measurements on rubberized and standard road surfaces and studied the interaction between texture and tyre/road noise. Loon et al. [27] gathered data from several countries in Europe on the age related performance of several types of road surfaces; they studied the mechanisms that cause the deterioration based on the spectral fingerprint of the wear processes. Licitra et al. [28] analyzed a complex interacting system that is composed of three main elements: pavement type, traffic loads, and climatic conditions. They then applied a new regression model to estimate the acoustic ageing of the investigated pavements.”

Point 4: Lines 41-44: I suggest replacing “seepage” with “drainage”.

Response 4: Thanks for your comments. “seepage” is revised as “drainage”.

Point 5: Section 2.1: Please specify the reference standard used to carry out the void fraction and connected void fraction values listed in Table 2.

Response 5: Thanks for your careful review and comments. The void fraction and connected void fraction were calculated according to the T-0708 volume method in the specifications for JTG E20–2011 [35].

Point 6: Table 2: Please insert the unit of measure of sieve size;I suggest replacing “Asphalt dosage” with “Asphalt content” everywhere.

Response 6:  Thanks for your careful review and comments. The unit of measure for the mesh size has been added, and “Asphalt dosage” is revised as “Asphalt content”.

Point 7: Line 139: The authors specify the name of the X-ray tomography CT digital analysis equipment v|tome|x s produced by General Electric Company. I suggest to avoid product or brand names and to use general terms as “X-ray tomography CT digital analysis equipment” specifying its characteristics.

Response 7: Thanks for your careful review and comments. It has been revised as “X-ray CT digital analysis equipment was used for the test.”

Point 8: Lines 155-167: This section is not well explained and organised. I suggest to better explain the sequence of the identification process of connected void fraction. Why are there eight values?

Response 8:  Thanks for your careful review and comments. The void is usually divided into three parts: connected void, semi connected void and closed void. Only the connected void is the channel for the migration of water and sound. In the two-dimensional image obtained by CT scanning, it is necessary to identify the void in the image layer by layer. As shown in the figure below, assumed void to be a value of 0, the non-void area is 1, when there is a void somewhere, identify the eight surrounding areas and observe whether there is a void. If it exists, it is a connected void, and it is continued to be identified. If it does not exist, it is a closed void, and the identification is stopped.

1

1

1

1

0

1

1

1

1

It has been revised as “The specific identification process of vertically (Z-axis) connected voids is assumes a value of 0 for a void. Each binary image is identified, and when the value is 0, the nearest eight values around it are checked. If one of the eight values is 0, all the values in the area with 0 are classified into one category. The void is considered to be connected in the area and is marked as the same void; the same steps are repeated until the last image has been checked. Finally, the images are individually checked from the first image at the top to the last image at the bottom, and all connected structures are defined as connected voids.”

Point 9: Lines 293-294-303-304-354-356: I suggest not to use formulas within the text.

Response 9: Agree. The formulas within the text has been deleted.

Point 10: Tables 11-12: Table 12 is repetitive. Please add the column of curvature values in table 11 and delete table 12.

Response 10: Thanks for your careful review and comments. It has been revised.

Table 12. Characteristic parameters of the connected void in EPA-13 using CT scanning.

Asphalt content (%)

Void fraction (%)

Connected void fraction (%)

Connected void volume (cm³)

Equivalent diameter (mm)

Surface area (mm²)

Curvature

5.5

13.24

11.72

44.36

43.92

134396.76

2.573

5.1

14.53

11.12

50.38

45.82

137436.09

2.439

4.7

15.82

14.44

55.92

47.44

139672.11

2.273

4.3

17.31

16.03

62.10

49.13

145212.58

2.077

Point 11: CONCLUSIONS: The conclusions are should be improved. Referring the discussion of results, the authors should emphasize the potential of the parameters obtained using CT scanning, as the possibility of predicting the sound absorption coefficient using the connected void fraction. Also, I suggest to put something about future developments at the end of the conclusions.

Response 11: Thanks for your careful review and comments. The conclusion has been revised as follows:

“In this study, the void characteristics and sound absorption characteristic parameters of epoxy porous asphalt mixture were tested using X-ray CT technology and an acoustic impedance tube. Their relationship was examined, and the following conclusions are drawn based on the findings of the study.

  1. The void characteristics of epoxy porous asphalt mixture obtained using CT scanning (meso scale) exhibited a good correlation with the measured values (macro scale). This indicated that the combination of CT scanning technology and digital image processing can not only characterize the complex meso morphological characteristics and spatial distribution of voids in asphalt mixture, but also accurately characterize the overall size of voids. This provides a good entry point to better analyze the void characteristics of the asphalt mixture.
  2. Both vertical and radial scanning demonstrate that the surface void fraction of the specimen is greater than the internal void fraction. The voids at the upper and lower parts are large, those in the middle part near the lower surface are relatively uniform, and those near the upper surface fluctuate significantly. The void fraction gradually decreases as the section approaches the axis. In addition, the number of voids in the EPA-13 epoxy porous asphalt mixture specimen is very high, and the closed voids below 1 mm³ account for more than 89% of all voids.
  3. The order of the grey correlation degree between the peak and average values of the sound absorption coefficient and the void characteristic parameters is as follows: connected void fraction > void fraction > equivalent diameter of connected void > surface area of connected void > curvature. Moreover, the correlation coefficient R² is very high. Therefore, the sound absorption coefficient can be predicted and analyzed based on the connected void fraction.
  4. In this study, the technical route was clear and feasible. In the next step, this technology can be used to study the drainage characteristics and road surface skid resistance of porous pavement.”

Reviewer 3 Report

The article by Xiaolong Li et al  reports a study on the relationship between void characteristics and sound absorption characteristics of epoxy porous asphalt mixture. The study was done on Marshall specimens by scanning and then reconstructing under different void fractions using X-ray CT technology and digital image processing method to obtain void characteristic parameters such as void fraction, connected void fraction, and curvature.   The results were then correlated to the sound  absorption coefficient  obtained at different frequencies using an acoustic impedance tube. The correlations between the parameters obtained by the two methods were then analyzed using  the grey correlation analysis. The  results showed that the void characteristics of epoxy porous asphalt mixture obtained by CT scanning correlated well with the measured values. It was found that the difference between the void fraction and connected void fraction gradually decreased with the increase in the void fraction. Radial and vertical scanning demonstrated that the surface void fraction of the specimen was greater than the internal void fraction and that the number of closed voids below 1 mm³ accounted for more than 89%. 

General remarks

This paper is interesting but a minor additional information to appreciate the limits of the impedance tube measurements is necessary.

Please give the lower and higher cut-off frequencies that are function of the length and the diameter of the impedance tube.

See Mohamed Ben Mansour et al, Influence of compaction pressure on the mechanical and acoustic properties of compacted earth blocks: An inverse multi-parameter,  Applied Acoustics 125 (2017) 128–135, http://dx.doi.org/10.1016/j.apacoust.2017.04.01 .

Please cite appropriately.
acoustic problem 

Author Response

Point 1: This paper is interesting but a minor additional information to appreciate the limits of the impedance tube measurements is necessary. Please give the lower and higher cut-off frequencies that are function of the length and the diameter of the impedance tube.

Response 1: Thanks for your careful review and comments. The acoustic impedance tube system consists of impedance tube, high-end microphone, data acquisition card, computer, etc. The pipe diameter is 100mm, the length is 90cm, and the measurement frequency range is 63 Hz ~ 1600 Hz. Relevant information has been improved in the text.

Point 2: See Mohamed Ben Mansour et al, Influence of compaction pressure on the mechanical and acoustic properties of compacted earth blocks: An inverse multi-parameter, Applied Acoustics 125 (2017) 128–135, http://dx.doi.org/10.1016/j.apacoust.2017.04.01.

Response 2: Thanks for your careful review and comments. The literature has been cited in the appropriate place in the text.

Round 2

Reviewer 1 Report

Thank you for addressing my comments.

Author Response

Thank you for your comments on our manuscript,those comments are all valuable and very helpful for improving the quality of our paper. 

Reviewer 2 Report

The authors did an excellent work in improving all the suggestions. The paper is almost ready for being publish, only a real fast minor check is suggest.

reference 6 is badly written. please check the authors.

avoid bullets in the conclusions.

Please, consider this paper to be mentioned in the introduction as a relevant new technique: Bianco, Francesco, et al. "Stabilization of a pu sensor mounted on a vehicle for measuring the acoustic impedance of road surfaces." Sensors 20.5 (2020): 1239.

Author Response

Point 1: Reference 6 is badly written. please check the authors.

Response 1: Thanks for your careful review and comments. It has been revised as “Krishnan et al. [6] used the developed framework to model the movement of voids in asphalt concrete, proposed voids filled with asphalt (VFA) as a parameter to model bleeding of asphalt concrete, and studied its variation for different loading and mixture condition.”

Point 2: Avoid bullets in the conclusions.

Response 2: Thanks for your careful review and comments. The conclusion has been revised as follows:

“In this study, the void characteristics of epoxy porous asphalt mixture obtained using CT scanning (meso scale) exhibited a good correlation with the measured values (macro scale). This indicated that the combination of CT scanning technology and digital image processing can not only characterize the complex meso morphological characteristics and spatial distribution of voids in asphalt mixture, but also accurately characterize the overall size of voids. This provides a good entry point to better analyze the void characteristics of the asphalt mixture. The order of the grey correlation degree between the peak and average values of the sound absorption coefficient and the void characteristic parameters is as follows: connected void fraction > void fraction > equivalent diameter of connected void > surface area of connected void > curvature, indicated the sound absorption coefficient can be predicted and analyzed based on the connected void fraction. In the next step, this technology can be used to study the drainage characteristics and road surface skid resistance of porous pavement.”

Point 3: Please, consider this paper to be mentioned in the introduction as a relevant new technique: Bianco, Francesco, et al. "Stabilization of a pu sensor mounted on a vehicle for measuring the acoustic impedance of road surfaces." Sensors 20.5 (2020): 1239.

Response 3: Thanks for your comments. The literature has been cited in the appropriate place.

This manuscript is a resubmission of an earlier submission. The following is a list of the peer review reports and author responses from that submission.